# Haline Convection within a Fresh-Saline Water Interface in a Stratified Coastal Aquifer Induced by Tide

**Elad Ben-Zur [1],\*, Haim Gvirtzman [1] and Eyal Shalev [2]**

[1] Institute of Earth Sciences, Edmond J. Safra Campus, The Hebrew University of Jerusalem, Givat Ram, Jerusalem 9190401, Israel; haim.gvirtzman@mail.huji.ac.il

[2] Geological Survey of Israel, 32 Yesha'ayahu Leibowitz St., Jerusalem 9692100, Israel; eyal@gsi.gov.il

\* Correspondence: elad.benzur@mail.huji.ac.il

**Abstract:** Sea-tide effects on the fresh-saline water interface (FSI) in a stratified coastal aquifer are examined through laboratory experiments. The physical model, a two-dimensional rectangular flow tank, is filled with layered aquifers and aquitards. The aquifers serve as the main entrances/exits of water to/from the system through significant horizontal flows, creating unstable conditions of heavier saline water above lighter freshwater for short periods of time. Several processes create mixing; this instability results in haline convection, creating downward fingering, stable rising of horizontal saltwater front, and unstable upward fingerings of flushing freshwater. The time lag between the sea tide fluctuations and the emergence of adequate fresh- and saltwater is higher in a stratified system compared to a homogeneous system. Furthermore, longer tide cycles lead to the enlargement of the FSI's toe horizontal movement range. The combination of tidal forcing with a layering aquifer structure leads to a wider FSI by creating a significant salt- and freshwater mixing inside each layer, vertical flows between the layers, and saltwater bodies at isolated areas. Haline convection within the FSI might be the reason for the wider fresh-saline interfaces that are found in field studies.

**Keywords:** sea tide; fresh-saline water mixing; groundwater level fluctuations; physical model

## 1. Introduction

Sea tide oscillations influence both the groundwater level (GWL) and the location of the fresh-saline interface (FSI) in coastal aquifers. They induce groundwater level fluctuations with a periodicity similar to tidal periodicity [1,2], as well as fluctuations of the FSI's location at the same periodicity [3–5]. Using a laboratory physical model, together with a matching numerical model, Levanon et al. [6] found a significant time lag between tide fluctuations and GWL and FSI responses. The time lag of the salinity values of groundwater within the FSI increases with depth and with distance from the sea floor. The tide effect is considered one of the major reasons for the wide mixing zone of the FSI found in field studies throughout the world. Oscillating vertical flows due to tide fluctuations greatly contribute to the mixing of fresh- and saltwater in coastal aquifers, and enhancement of FSI thicknesses [7,8].

Another major reason for the wider FSI is aquifer stratification. Most coastal aquifers have a complex heterogeneous and anisotropic nature [9]. This is a result of the changing sedimentary depositional environments and the presence of karst, fractures, and dykes [10]. Rumer and Shiau [11] described the shape of the FSI in a layered coastal aquifer and showed that the FSI is refracted upon passing between adjacent layers. The FSI slope in layers characterized by a high hydraulic conductivity is relatively moderate compared to layers with lower hydraulic conductivity, whose slope is steeper. Mualem and Bear [12] considered a thin horizontal semi-permeable layer dividing two homogeneous parts of a coastal aquifer. Using field observations and laboratory experiments, they demonstrated that under these conditions, a discontinuity in the shape of the interface occurs, such that it

is divided into two parts, below and above the semi-permeable layer. Bakker [13] found that the FSI toe location in a semi-confined aquifer with a leaky layer depends on several dimensionless parameters, including the hydraulic gradient, density difference between the two water bodies, horizontal and vertical hydraulic conductivities of the layers, the length and thickness of the leaky layer.

The FSI was found to be wider than expected in many coastal aquifers, up to the order of tens of kilometers [14,15]. Beside the tide and stratification, several physical parameters have been offered to explain the occurrence of the wider mixing zone: transverse and longitudinal dispersion and non-uniform advection, the aquifer's heterogeneity and anisotropy, and temporal changing of groundwater or sea levels. Furthermore, laboratory and numerical studies demonstrated that heterogeneous hydraulic conductivities lead to spatially varying preferential flows and thus to an ununiformed advection [16]. Lu et al. [17] found that when a low-conductivity layer overlies a relatively high-conductivity layer, the mixing zone in the upper layer is thickened due to enhanced separation of the freshwater–saltwater mixture flow. In addition, the aquifer's heterogeneity may support seawater penetration along preferential flow paths and induces unstable density configurations, creating saltwater fingers and thus a thicker FSI [18]. Nevertheless, each factor has only a moderate effect, and the factors are not yet well understood [18].

The use of physical models provides an ideal setup for conceptualization, visualization, and measurement of groundwater flow and solute transport through an aquifer [19–22]. The impact of stratified heterogeneity has been studied on different phenomena [23–26]. In this study, we used such a physical model, demonstrating the effects of tide and stratification on the FSI width. We demonstrate for the first time that haline convection within the FSI plays a major role in the mixing process at coastal aquifers. Our laboratory experiments compare homogeneous versus stratified media and analyze different boundary conditions and tide frequencies. Our major objective is visualization and characterization of the new mixing mechanism, but its numerical simulation is out of the scope of this study.

## 2. Methods

### 2.1. The Laboratory Model

The two-dimensional vertical laboratory model simulates a stratified coastal aquifer that is affected by sea-tide, with a high-resolution monitoring ability at the FSI. The model is divided into three distinct chambers (Figure 1): a central flow chamber, which serves as an aquifer and contains the porous medium, and two side chambers filled with water, which define the boundary conditions during the experiments. The left chamber represents the saltwater boundary at the seaside, and the right chamber represents the inland freshwater boundary of the regional aquifer. These side chambers are separated from the main chamber by a fine net that prevents the passage of granular material. The central chamber is 100 cm long, 50 cm high, and 3 cm wide. The narrow width of the tank ensures that the problem examined could be defined as two-dimensional, with negligible along-shore processes. For simulating a stratified aquifer, the tank is filled with two types of sand that are set in a layered structure: coarse sand with a grain diameter of 850–1500 μm, and fine sand with a grain diameter of 100–350 μm. The thickness of the layers is about 5 cm. In reality, the horizontal and vertical intrusion of the fresh-saline interface is at the scale of hundreds of meters, while the thickness is in the range of tens of meters. The porous media of homogeneous aquifer experiments is composed only of coarse sand. The sand is washed with hydrochloric acid to remove the oxide coatings of the quartz grains. The freshwater in the experiment is colorless tap water, and the saline water is prepared by dissolving NaCl in tap water and adding commercial red food dye. The density of the saline water is 1027 kg m$^{-3}$.

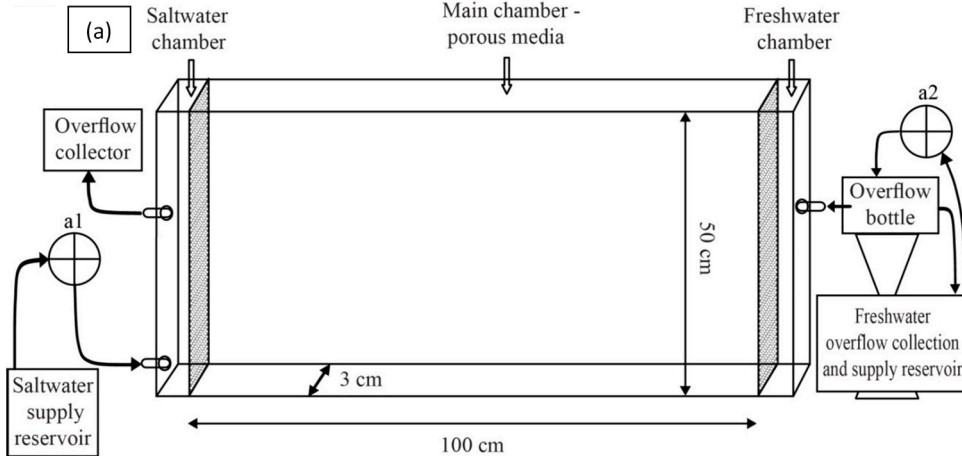

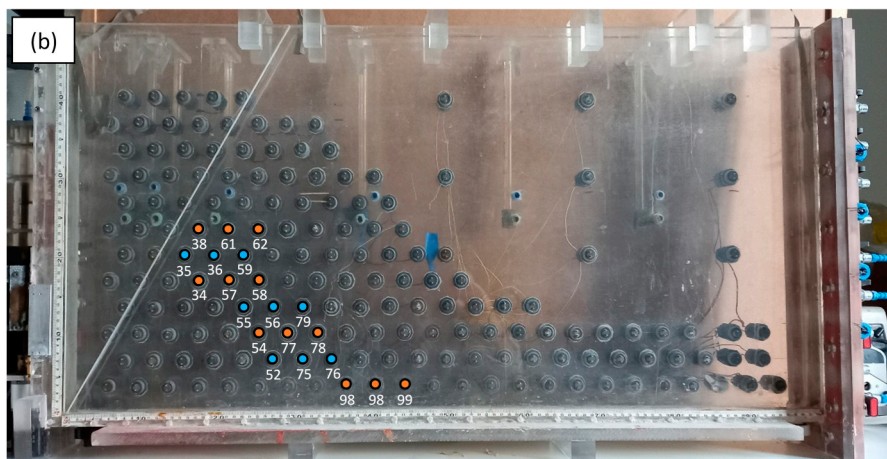

**Figure 1.** (**a**) A schematic description of the physical model. (**b**) A photo of the physical model and sensors. The colored (and numbered) sensors are those whose results are detailed in the text.

The initial and boundary conditions of the experiment are defined by the water levels at the side chambers. At the right boundary, freshwater is pumped from a supply reservoir into an overflow bottle connected to the right chamber. At the left boundary, the saltwater is pumped from another supply reservoir into the left chamber. The water level at the right boundary remains constant throughout the experiment and is higher than at the left boundary. This forms an initial hydraulic gradient, and the water flows through the porous medium from right to left, simulating a seaward groundwater flow. The overflow at the left boundary is diluted seawater as a result of mixing between the freshwater and the saltwater. The overflow pipe at the left chamber is connected to a computer-controlled engine that enables quasi-tidal fluctuations of the saltwater at the left boundary, which defines the boundary condition at the seaside. The amplitude and the wavelength of the sinusoidal tidal wave are definable by setting time-serious sublevels. The tidal wave used in the experiments is harmonic with a constant amplitude [2]:

$$h\,(t) = h_0 + A\,sin(\omega t) \tag{1}$$

where $h(t)$ [L] is the left hydraulic head as a function of time, $t$ [T]; $h_0$ [L] is the initial hydraulic head without tide oscillation; $A$ [L] is the tide amplitude; and $\omega = 2\pi/T$, where $\omega$ [Rad T$^{-1}$] is the tidal angular frequency and $T$ [T] is the tide period. $A$ and $h_0$ are constants through all of the experiments, and their values are 1.5 cm and 30 cm, respectively.

The monitoring setup includes several measurement systems [6,20,21]. More than 150 sensors are placed at the backside of the flow tank for in situ voltage values (Figure 1b). The sensors were calibrated using several saline solutions with various electrical conductivities (Supplementary Material). These values are proportional to the density of the water, which varies between 997 and 1030 kg m$^{-3}$. Thus, high and low EC values in a sensor within the FSI area imply the existence of salt- and freshwater, respectively. The data from the sensors are collected at a rate of 2 sample per minute. Water level in both side chambers is measured using a Keller–Druck 33 PA transmitter with a set range of 0–0.1 MPa. The left transmitter also allows control of the 1.5-cm amplitude made by the engine. A barometric transmitter is used to calculate the actual water pressure at the side chambers during the experiment. The experiment is documented by a time-lapse camera. The different water bodies are visually apparent in the photos because the saltwater is dyed red.

### 2.2. Cross-Correlation Analysis

The results of the laboratory experiments, namely the EC values from the sensors and the left chamber's water level from the pressure meter, are received in the form of wave functions. In order to calculate the time lag between them, cross-correlation analysis is used. This analysis allows determining the similarity between two functions, $f(t)$ and $j(t)$, by shifting $j(t)$ along the time axis in constant time steps, represented by the variable $\tau$. At any time step, the level of correlation is being checked by calculating the convolution value, namely the integral of the product of the two functions, while one is shifted:

$$(f * j)(t) = \int_{\infty}^{\infty} f(t)j(t - \tau)d\tau \tag{2}$$

A high convolution value represents high similarity between the functions and the positive product of the aligned peaks. Thus, when this value is the maximum, the correlation is the highest, and the accepted $\tau$ is the requested time lag. The resulted time lags are those with a correlation value higher than 70%.

### 2.3. Experiment Setup

Several successful experiments were conducted, and each differed in one or more of the following parameters: left boundary profile, porous media structure, and the tidal period time (Table 1). The three key experiments are E1-60, E2-120, and E3-180, and they are characterized by an oblique left boundary and a layered heterogeneous porous media. They have different tidal periods of 60, 120, and 180 min, respectively, as mentioned in their ID. The ID of the other experiments also represent a change in one parameter: vertical left boundary (a cliff-like shore) in E4-VB-60 and E5-VB-180, and homogeneous aquifer in E6-Hm-60 and E7-Hm-180.

In all experiments, fresh- and saltwater flow constantly into the right and left chamber, respectively, and after a stabilized FSI is generated, the tide engine starts to act and changes the left hydraulic head. The tidal cycle starts and ends at the mean left level, 30 cm, and starts with the rising of the left level to the maximum high tide, according to a simple sine wave. The tidal waves are different only due to the different periods of time, as the amplitude of the left hydraulic head remains at 28.5–31.5 cm, and the experiments last for at least 3 tidal cycles. The right hydraulic head remains constant throughout the experiment, though minor fluctuations occur at the right as a response to changes in the left level.

All stratified experiments have the same layering structure. The lowest layer is made of coarse sand with a thickness of 5 cm. The layer above is a 3-cm-thick fine sand layer. The alternating coarse and fine layers repeat twice more each, and the uppermost layer is coarse. The thickness of the uppermost layer is larger, as it contains most of the unsaturated zone. The layers are numbered from 1 at the bottom to 7 at the top. In order to distinguish the layer types, coarse or fine, the layers are named after their type and number. For example, the lowest layer is named C1, and the layer above it is F2.

**Table 1.** The experiments details. At the experiment's ID, <u>VB</u> is Vertical Boundary and <u>Hm</u> is homogeneous. The number in each name represents the duration, in minutes, of the tide period. / and | describe oblique and vertical left boundary, respectively.

| Experiment ID | Left Boundary | Medium | Right Head (cm) | Period Time (min) | Video |
|---|---|---|---|---|---|
| E1-60 | / | stratified | 31.5 | 60 | Supplementary 1 |
| E2-120 | / | stratified | 32 | 120 | Supplementary 2 |
| E3-180 | / | stratified | 31.7 | 180 | Supplementary 3 |
| E4-VB-60 | \| | stratified | 31.5 | 60 | Supplementary 4 |
| E5-VB-180 | \| | stratified | 31.9 | 180 | Supplementary 5 |
| E6-Hm-60 | / | Homogeneous | 31.8 | 60 | Supplementary 6 |
| E7-Hm-180 | / | Homogeneous | 32 | 180 | Supplementary 7 |

## 3. Results

### 3.1. General Flow Patterns

Results of the homogeneous experiments show a well-defined continuous FSI, whereas in the stratified experiments, the distribution of the water bodies is very complex (Figure 2 and the seven supplementary video files), as follows. The two types of layers composed of coarse and fine sands have different hydraulic conductivities, reflected by a faster horizontal water flow through the coarse sand layers. The whole medium is divided into seven layers; the fine sand layers act as aquitards and the coarse sand layers function as semi-confined aquifers. The FSI is divided into seven smaller interfaces; each is determined by its specific characters and by the interaction with the bordering layers.

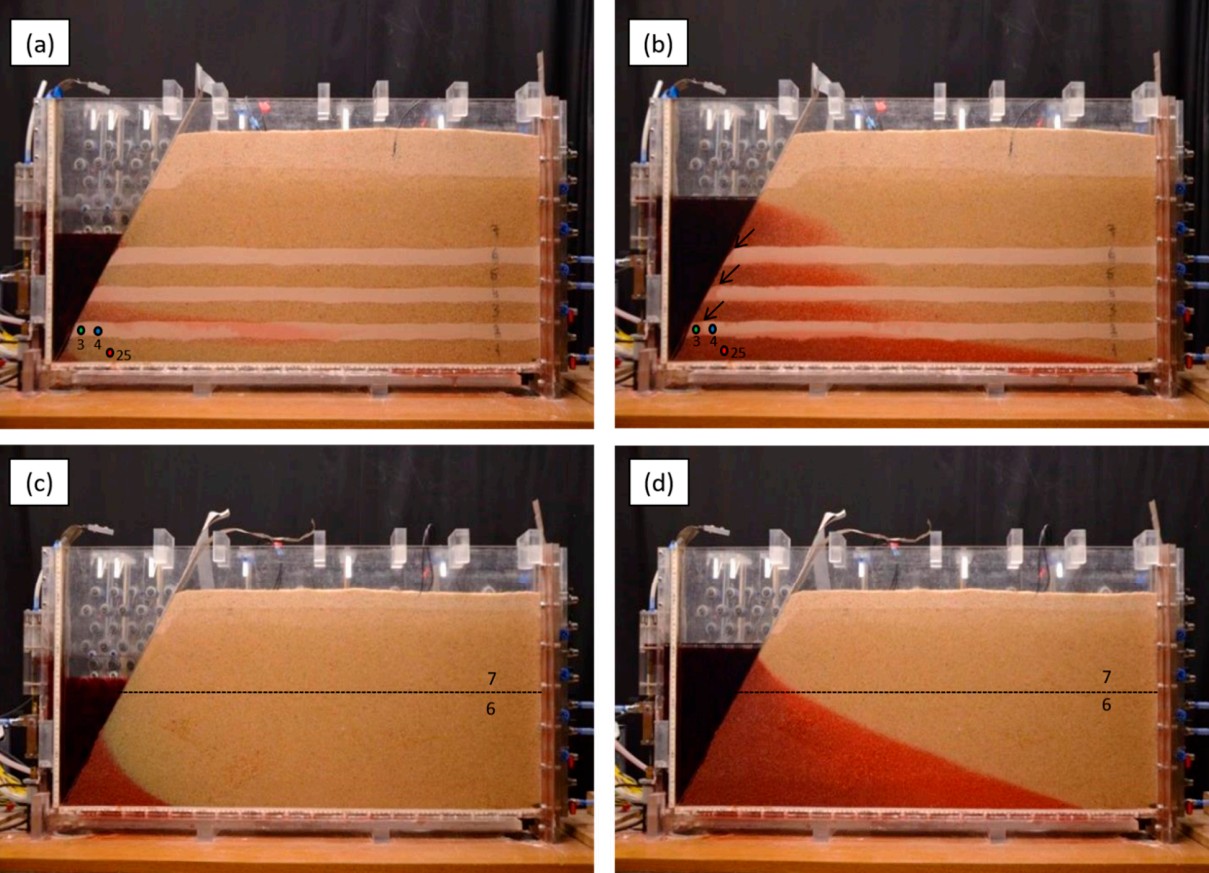

**Figure 2.** Saltwater is red and freshwater is colorless. Experiments E3-180 at low tide (**a**) and high tide (**b**), and E7-Hm-180 at low tide (**c**) and high tide (**d**). The arrows in (**b**) represent the maximum saltwater horizontal intrusion into the aquitards. The dashed line represents the height of layers 6–7 boundary as a reference.

There is a direct influence of changing sea level (left boundary) on horizontal water flow at the aquifer layers. While the sea level rises, saltwater intrudes tens of centimeters 'landward', pushing the freshwater away (Figure 2b and supplementary file #3). The same process repeats when the sea level decreases, as the saltwater retreats toward the left boundary (Figure 2a). The response is relatively fast—though not immediate—up to a few dozen minutes.

The rising of the left level also causes direct horizontal flow at the aquitard layers, though only up to a few centimeters, close to the left boundary (arrows in Figure 2b). Instead, the major water mass within the aquitards is pushed vertically by water from the aquifers below. Because the general flow direction in the entire system is toward the upper-left boundary, representing the drainage zone of a coastal aquifer toward the sea, it causes upward water flow from the aquifers to the above aquitards.

These flow patterns are also reflected by the relative EC values of adjacent sensors. At E3-180 (Figure 3), in response to the rising of the left level, saltwater flow arrives first to sensor 25, located at the aquifer C1. It then reaches sensor 3, which is located significantly closer to the saltwater origin, though placed in aquitard F2. After a much longer time, about 100 min, it reaches sensor 4, which is located only a few centimeters landward. This phenomenon repeats in the other sensors located throughout the mixing zone in the other stratified experiments.

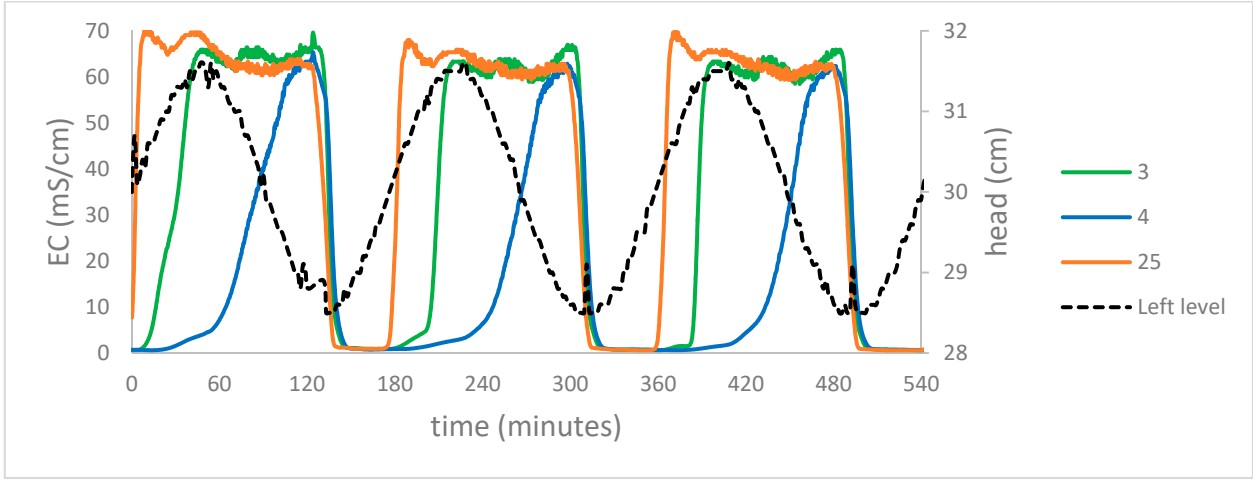

**Figure 3.** Left level head boundary conditions and water electrical conductivity at three sensors during E3-180. Sensor locations are shown in Figure 2a.

### 3.2. FSI's Location

The horizontal movement of the FSI toe between high tide and low tide may serve as a characteristic indicator of the tide effect. Indeed, comparing the stratified to the homogeneous experiments, for example E3-180 and E7-Hm-180 (Figure 2 and supplementary files #3 and #7), there is an approximately 20-cm difference in the horizontal range of the FSI's toe movement. Furthermore, among the stratified experiments, the horizontal movement of the FSI is characterized by two means. First, at the lowest aquifer, C1, the FSI's toe horizontal amplitude between high tide and low tide is the greatest (Figure 2a,b). Second, the amplitude of the toe horizontal movement increases with increasing tidal wavelength (E1-60, E2-120, and E3-180 and supplementary files #1, #2 and #3).

### 3.3. Vertical Flows

Figure 4 depicts in detail an illustrative experiment, namely E3-180. It includes six snapshots during one tidal cycle, each showing the distribution of the red saltwater and the colorless freshwater in the laboratory model at a specific time step. This experiment was chosen for describing the involved processes because it spanned over a longer tidal

wave (180 min), which allowed the processes to evolve and be visually expressed more successfully. The exact times of the six snapshots during the tidal wave are marked in Figure 4g.

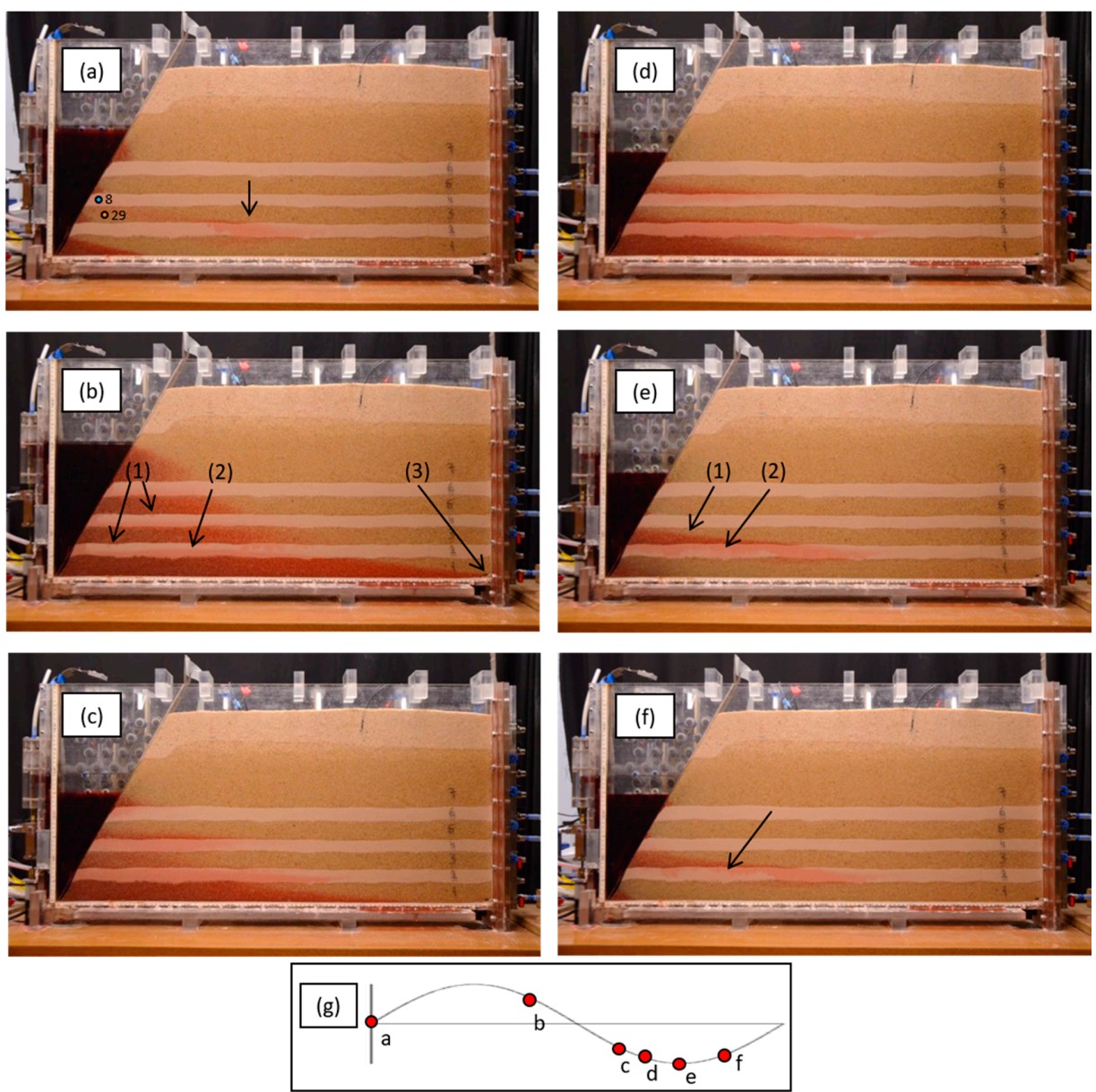

**Figure 4.** A tidal cycle in E3-180. Several phenomena are depicted, including a remaining saline pocket (**a**); salt fingers (**b1**); flat salt front (**b2**); maximum FSI's toe intrusion (**b3**); saltwater drainage horizontally from the aquifers (**c**); flooding from vertical saltwater again (**d,e1**); and flushing fingers (**e2,f**). Step timing is described in the schematic graph below (**g**).

At the beginning of the tidal cycle (Figure 4a), the FSI is divided into four distinct parts, located at the left edges of the four aquifers: C1, C3, C5, and C7. The rising head at the left boundary induces fast saltwater intrusion through these four aquifers, compared to the slow negligible flows at the aquitards (Figure 4b). However, the maximal intrusion occurs slightly after the high tide, as marked by point b in Figure 4g, because no steady-state conditions are achieved at the instantaneous high-stand. The next three snapshots (Figure 4c–e) describe the retreating of the saltwater due to the enforced flushing of fresh groundwater, enhanced by the dropping head at the left boundary. The retreating saltwater is expressed again by fast horizontal flow in the above-mentioned four aquifers. Here again, the maximal retreating occurs slightly after the low stand, as marked by point f in

Figure 4g, because of the unsteady conditions. The end of this tidal wave is described in Figure 4a, which is the beginning of the next wave.

The six snapshots in Figure 4 describe several additional important processes, namely three types of vertical flows: (1) steady upward-moving front of saltwater from an aquifer into an aquitard, (2) unsteady downward fingering of saltwater from an aquifer into an aquitard, and (3) unsteady upward fingering of freshwater from an aquifer into an aquitard. It should be noted that the second and the third mechanisms are haline convections induced by density difference. In addition, saline water pockets are sometimes created. These phenomena are described below.

A steady upward moving front of saltwater from an aquifer into an aquitard (for example, 2 in Figure 4b) occurs during high tide, either before or after its maxima, due to horizontal intrusion of saltwater through aquifer C1, thereby pushing the freshwater in F2 upward. Because saltwater in the aquifer C1 is denser and heavier than the freshwater in F2, it is physically stable (lighter water above heavier water), thus an upward-moving front takes place.

An unsteady downward fingering of saltwater from an aquifer into an aquitard (1 in Figure 4b) also occurs during high tide. At this time, for a short duration, intruding heavy saltwater is placed in aquifers C3 and C5 atop light freshwater in aquitards F2 and F4, respectively, which creates unstable conditions. As a result, downward fingering is initiated, yet it is very small due to the low hydraulic conductivity of the aquitards and the short duration.

An unsteady upward fingering of freshwater from an aquifer into an aquitard (2 in Figure 4e,f) occurs during low tide, either before or after its minima, due to seaward horizontal flushing of freshwater through aquifer C1, thereby pushing the saline water in F2 upward. Because freshwater in the aquifer C1 is lighter than the saline water in F2, it is physically unstable, thus an upward fingering takes place.

Sometimes, the upward-flushing freshwater fingers originating from C1 (Figure 4f) dilute the saltwater in F2 and split the saline water body into several smaller bodies, thereby creating a saline pocket (Figure 4a and supplementary files #3 and #5). Saline pockets are not created in F4 and F6 due to the shorter duration available for the flooding by saline water and re-flushing by freshwater. When a diluted saline water pocket within an aquitard is flushed away by freshwater, it creates a saline flow pulse within the adjacent aquifer, regardless of the tide periodicity. Such a phenomenon is indicated by the sensors and is seen in Figure 5. In principle, the EC variations at most sensors fit a sinusoidal tidal wave, with a time lag compared to the head variations at the left-side boundary (Figure 3). However, when a saline pocket is washed out, the EC variations are a combination of the primary sinusoidal wave and a secondary saline pulse, for example, as seen at the two sensors in Figure 5. The magnitude and duration of the secondary pulse are obviously smaller.

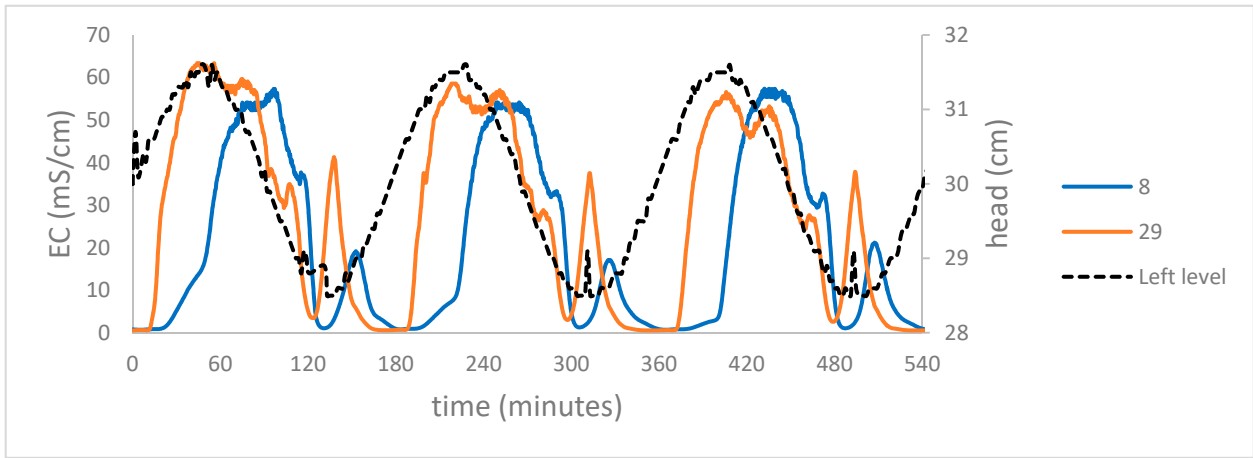

**Figure 5.** Head oscillation at the left boundary and electrical conductivity variation of groundwater at two sensors during E3-180. Sensor locations are shown in Figure 4a.

### 3.4. Time Lags

The time lag at a specific point is calculated by the cross-correlation analysis between left boundary head oscillations and EC oscillations in the adjacent sensor. Sensors that do not demonstrate a sinusoidal EC behavior are characterized by a lower correlation (<0.7), and thus are not included. They are exceptional because of their far locations from the FSI, the secondary saltwater flows, or their location within a saline pocket.

Results show the time lag in stratified systems is generally higher than in homogeneous ones by a factor of two. In a homogeneous porous medium, the time lag increases with distance from the drainage point, in the vertical direction as well as in the horizontal one, as shown in previous studies [6]. Results also show (Figure 6) that, in the stratified system, the time lag is significantly higher at the aquitards than those at the aquifers. Furthermore, time lag increases downward among both the aquifers and aquitards.

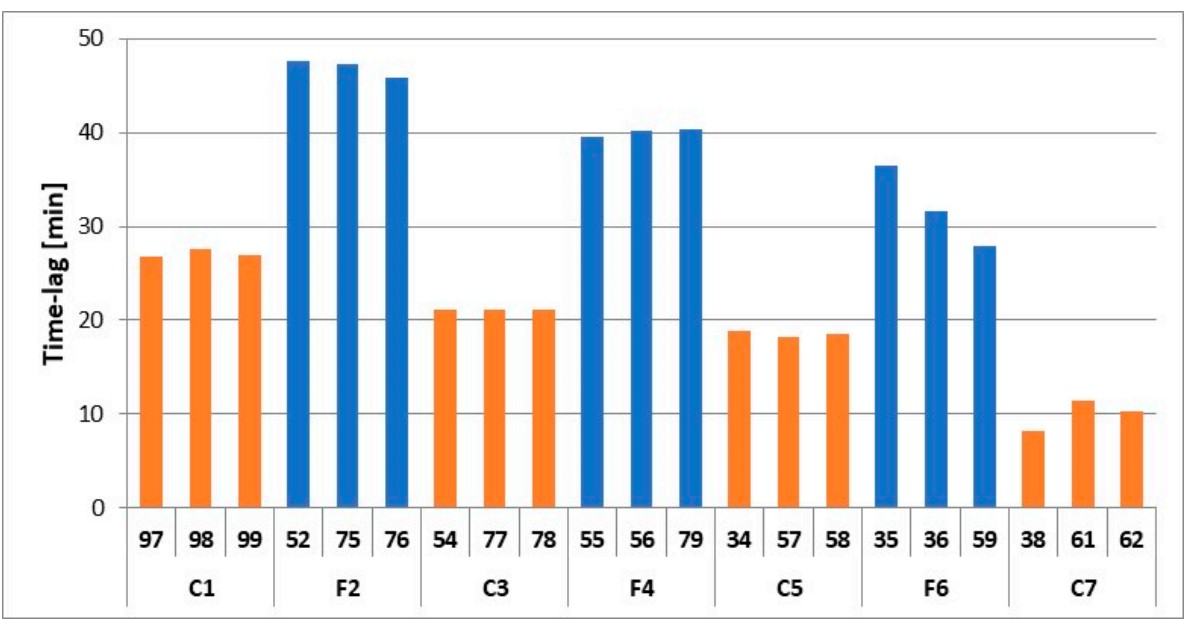

**Figure 6.** Time lags of EC oscillations at several sensors relative to head oscillations at the boundary in E2-120. Sensor locations are shown in Figure 1.

## 4. Discussion

The objective of this study was to visually demonstrate a new mechanism, haline convection, which takes place within the FSI. This mechanism is probably the one that most affects the expansion and widening of the mixing zone in actual coastal aquifers. To the best of our knowledge, this phenomenon has never been visualized and published. However, this study lacks numerical simulations, which would help in quantifying and verifying the hypothesis. Nevertheless, the controlled laboratory experiments, the repetition of the results at various boundary conditions, and the visual results by means of the supplementary video files make the results compelling.

### 4.1. The Flow Field

Unlike homogeneous aquifers, where all groundwater flow vectors point toward the shoreline (Figure 2c,d), horizontal flow is dominant in stratified aquifers and vertical flow is dominant in stratified aquitards (Figure 2a,b). This difference affects the drainage location of the fresh groundwater at the seashore, namely the submarine groundwater discharge (SGD). Under homogeneous aquifer conditions, the SGD is quite narrow and adjacent to the shoreline. However, under stratified aquifer conditions, the SGD expands significantly because the horizontal flow in the aquifers, in between the aquitards, forces the discharge seaward at a deeper depth.

Because the sea-side boundary enforces tide oscillation, no steady-state flow and transport condition prevail throughout the experiment. However, as the changes at the boundary are smaller and slower, the chances for developing quasi–steady-state conditions increase. In these conditions, transport processes have time to adjust to developing hydraulics. This phenomenon is observed by the FSI toe horizontal movement. As the tide-cycle duration increases, the chances for quasi–steady-state conditions increase, thus the toe horizontal range of movement increases.

### 4.2. Mixing Mechanisms

The total thickness of the FSI in coastal aquifers depends on the mixing mechanism. Several processes create mixing: limited horizontal flow in aquitards, rising of flat salt front into a layer saturated with freshwater, haline convection that forms saline water fingers, and haline convection that forms freshwater fingers (Figure 7). The fast horizontal water flow in the aquifers compared to the restricted flow in the aquitards seems to be the major mixing mechanism. Saline water pockets that are trapped due to partial flushing by freshwater seem to be the second important mixing mechanism. This partial flushing is the result of haline convection within the FSI.

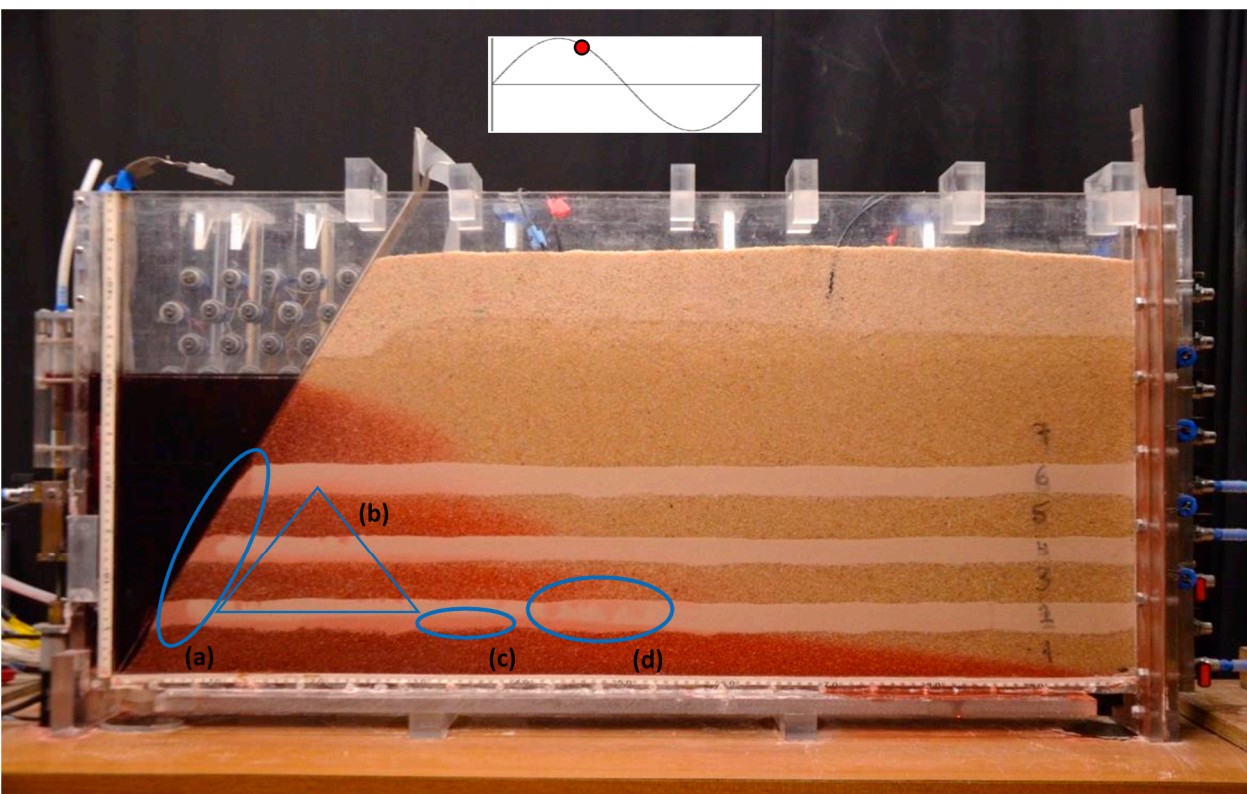

**Figure 7.** Several mixing mechanisms at the aquitards during high tide, E3-180: (**a**) restricted horizontal flow; (**b**) haline convection, namely downward salt fingers from the above aquifers; (**c**) upward salty front; (**d**) flushing of freshwater fingers into saline pockets. This snapshot was taken slightly after the high tide (box at the top), when the seawater intrusion is maximal.

### 4.3. Hydraulic Conductivity

The main parameter that controls the flow field and the mixing mechanisms is the hydraulic conductivity. The layered configuration of the system with different hydraulic conductivities creates complicated density distribution and unstable conditions that lead to several mixing processes. Different hydraulic conductivities of the layers will enhance or reduce mixing. Furthermore, within a layer, the unstable conditions of saline-dense water overlying freshwater create haline convection that depends on the hydraulic conductivity.

Convection is characterized using the dimensionless Rayleigh number that linearly depends on the hydraulic conductivity. Higher conductivity within a layer will increase convection and mixing.

Another important factor is the thickness of the layers. Thick layers encourage haline convection and mixing. The density and viscosity of seawater do not vary in orders of magnitude and may be considered as constant.

## 5. Summary

Tidal effects on the FSI in a stratified coastal aquifer were analyzed using laboratory experiments. It was found that the combination of tidal forcing with a layering structure leads to a wider FSI. A new mixing mechanism at the FSI was hypothesized and visually demonstrated, namely haline convection. This mechanism might be the major reason for wider FSIs that are found in field studies. The following processes were observed:

1. The faster horizontal flow through aquifers leads to an intense horizontal intrusion of saline water and intense flushing by fresh groundwater during high tide and low tide, respectively, thereby expanding the mixing zone;
2. The preferential flow through aquifers creates unstable conditions where dense saline water is placed above light freshwater for short-time periods. Haline convection is initiated, further intensifying the mixing process;
3. The overall time lag of the density changes within the FSI area, as reflected by EC values, in response to tidal fluctuations in stratified systems that are higher compared to homogeneous systems;
4. The periodicity of the density value in the aquifer follows the periodicity of the tide (with a time lag); yet it may be disturbed by secondary flows of flushing saline water pockets;
5. The duration of the tidal cycle has a major effect on the establishment of quasi–steady-state conditions, especially on a stratified system, by means of the enlargement of the FSI's toe horizontal movement range;
6. The thickness of the FSI in a stratified aquifer will be wider than in a homogenous aquifer.

**Supplementary Materials:** The following are available online at https://www.mdpi.com/article/10.3390/w13131780/s1, Figure S1: Calibration of the density sensors used in the experiment. Seven supplementary video files of the experiments.

**Author Contributions:** Conceptualization, H.G. and E.S.; Funding acquisition, H.G.; Investigation, E.B.-Z.; Methodology, H.G., E.S. and E.B.-Z.; Supervision, H.G. and E.S.; Writing—original draft, E.B.-Z.; Writing—review & editing, H.G. and E.S. All authors have read and agreed to the published version of the manuscript.

**Funding:** This study was financed by the Israeli Water Authority.

**Institutional Review Board Statement:** Not applicable.

**Informed Consent Statement:** Not applicable.

**Data Availability Statement:** All data are available on request at the following e-mail address: elad.benzur@ mail.huji.ac.il.

**Acknowledgments:** We would like to thank Elad Levanon, Yehuda Levi, Yosi Sherer, Arik Diament, Itzhik Simchayof, and Andrey Pudelko for their help in the laboratory. We also thank Benjamin Snow, ELS, MWC for editing this manuscript.

**Conflicts of Interest:** The authors declare no conflict of interest.

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
