# Peer review of "Haline Convection within a Fresh-Saline Water Interface in a Stratified Coastal Aquifer Induced by Tide"

_water, doi:10.3390/w13131780_

Round 1

Reviewer 1 Report

In my previous Revision, one of the points, a strong one, concerned salinity, a key parameter for the work presented in the manuscript (e.g. salinity fluctuations, salinity peaks, salinity oscillations, salinity behavior, salinity pocket,  salinity pulse, periodicity of salinity). The authors corrected the units in which the quantity is expressed, nevertheless, against my expectations, they did not demonstrate adequate knowledge or sensitivity to meaning of the parameter, namely  to the measurement  method and procedures and calibration.

Indeed the title of the manuscript refers “…a fresh-saline water interface….” and the term salinity appears  15 times,  without a definition and without a proper description of the method, equipment and experimental procedure:

Abstract - 1 time

Introduction – 2 times

Methods-  2 times

Lines 112-114- “…168 electrodes placed  at the backside of the flow tank for in-situ electrical conductivity (Figure 1b), which are  proportional to the salinity of the water. The data from the electrodes is measured …”.

R: a) “Electrodes”- The standard procedure for measurement of salinity is based on electrochemical cells (each cell has two electrodes);

  1. b) “situ electrical conductivity which are proportional”- Conductivity values are proportional.

Line 128- “… conductivity values  from the electrodes (which is proportional to salinity)”; proportionality has already been stated at line 113.

R: The conductivity values pertain to the  cells.

Results- 8 times

Line 187- These flow patterns are also reflected by the relative electrical conductivity (proportional to salinity) values

R: Proportional /repeated

Line 239

Lines 261, 26, 270

Lines 278, 279

Line 291- “… salinity oscillations at several electrodes”

R: Electrodes????

Discussion-  Strangely ABSENT!

Summary

Lines 346, 348.

Author Response

Comment: In my previous Revision, one of the points, a strong one, concerned salinity, a key parameter for the work presented in the manuscript (e.g. salinity fluctuations, salinity peaks, salinity oscillations, salinity behavior, salinity pocket, salinity pulse, periodicity of salinity). The authors corrected the units in which the quantity is expressed, nevertheless, against my expectations, they did not demonstrate adequate knowledge or sensitivity to meaning of the parameter, namely to the measurement method and procedures and calibration.

Indeed the title of the manuscript refers “…a fresh-saline water interface….” and the term salinity appears 15 times, without a definition and without a proper description of the method, equipment and experimental procedure:

Answer:  Indeed, our main measurement gauges are not electrodes as we first named them, and it would rather be better to name them ‘sensors’. We present the results of these sensors in EC values, which reflect better the degree of the water salinity. We have elaborated the explanation of the calibration of the sensors in the ‘Methods’.

The term ‘salinity’ was used in several meanings and now we have changed it:

  1. When it was used to describe a salty object it was replaced by the term ‘saline’ (e.g. ‘saline pocket’).
  2. When we demonstrate direct results we replaced it with ‘EC values’, as these are the actual results.
  3. When we wanted to describe the process of water with different salinities that flows within the FSI area, we simply use ‘salinity’. It repeats only a few times in the paper.

Reviewer 2 Report

This paper analyze trough a physical model the interface dynamics of a salt water intrusion into a coastal aquifer. The modelling setup is constituted by a sandbox equipped with probes for the salt concentration indirect assessment, while a camera allows the visualization of the salt water marked with a dye.

This work is in line with previous studies of the same authors, with significant differences.

The topic is interesting and deserves attention from the scientific community as demonstrated by the body of literature dealing with the study of coastal aquifers.

The paper is well written and well organized, figures are clear and high quality.

In the sequel my main comments

  • A1 - First of all, my main concern is about the innovative contribution of this work, I notice a lack of quantitative results, the visual analysis of the flow should be supported by quantitative metrics associated to the phenomena observed.
  • A2- The only parameter quantified here is the time lag between the tide height and the salinity, (and in this case why the assessment of the time lag is not performed by a simple cross correlation ?)
  • A3 - A more detailed discussion of the effects of other parameters is required (see comment 3)

  • B1 - My second concern is about the physical model: the experimental investigation should by based on a proper choice of the geometric dimension and materials in order to obtain similarities of significant dimensionless parameters. It is not clear in the text if Authors adopt similarities; this is a key issue in the physical investigations and requires a detailed explanation in the text.

  • C1- My third concern is about the generality of the result:

different aquifers (homogeneous, free surface vs heterogeneous, confined) were compared, can the different flow dynamics support the results explaining the differences on the FSI?

  • C2-The dynamics of the observed phenomena strictly depend on the tidal period, but a significant effect is played also by the hydraulic conductivity which rules, together with the head gradient and the salinity gradient the flow velocity. The key role of K should be clearly emphasized and discussed in the text, and the theoretical discussion of the effect of K on the FSI and on the lag time should be clarified.

Finally I suggest the “reshape” of the results emphasizing the most innovative aspects e summarizing other points

In my opinion, the paper can be considered for publication after major revision

Author Response

Comment: A1 - First of all, my main concern is about the innovative contribution of this work, I notice a lack of quantitative results, the visual analysis of the flow should be supported by quantitative metrics associated to the phenomena observed.

Answer: In this paper, the time lag is quantified using cross correlation. Other parameters are difficult to measure in the laboratory and can only be examined using a numerical model. Indeed, some of the main finding of this work, such as mixing mechanisms, are presented without quantitative analysis. Nevertheless, in our opinion, these mechanisms were not considered previously. Our results show new mechanisms for mixing in coastal aquifers.

Comment: A2- The only parameter quantified here is the time lag between the tide height and the salinity, (and in this case why the assessment of the time lag is not performed by a simple cross correlation ?)

Answer: The time-lag is calculated by the cross-correlation method, and presented in a form that should ease the reader to distinguish the differences between the two layers type. See sections 2.2 and 3.4.

Comment: A3 - A more detailed discussion of the effects of other parameters is required (see comment 3)

 Answer: We have added a new paragraph (4.3) at the end of the discussion on the importance of the hydraulic conductivity value.

Comment: B1 - My second concern is about the physical model: the experimental investigation should by based on a proper choice of the geometric dimension and materials in order to obtain similarities of significant dimensionless parameters. It is not clear in the text if Authors adopt similarities; this is a key issue in the physical investigations and requires a detailed explanation in the text.

 Answer: The physical model size is 100 X 50 cm. Layers’ thickness is ~5cm. In reality, the fresh-saline interface horizontal and vertical intrusion is at the scale of hundreds of meters. Layers’ thickness is up to tens of meters. The relative lengths of all dimensions are similar. We now explain this in the Method section.

Comment: C1- My third concern is about the generality of the result: different aquifers (homogeneous, free surface vs heterogeneous, confined) were compared, can the different flow dynamics support the results explaining the differences on the FSI?

Answer: Yes, thank you for this comment; this is an important outcome of this work. The thickness of the FSI in a heterogeneous aquifer will be wider than in a homogenous aquifer. We added this to the summary.

Comment: C2-The dynamics of the observed phenomena strictly depend on the tidal period, but a significant effect is played also by the hydraulic conductivity which rules, together with the head gradient and the salinity gradient the flow velocity. The key role of K should be clearly emphasized and discussed in the text, and the theoretical discussion of the effect of K on the FSI and on the lag time should be clarified.

Answer: We have added a new paragraph (4.3) at the end of the discussion on the importance of the hydraulic conductivity value.

Reviewer 3 Report

It is rather difficult reading this manuscript as the figures provided are not clear. Many important features such as salt fingers and flushing fingers can not be seen(on my screen). Yet, the supplementary films are of much higher quality and show these features clearly. The authors should choose figures more carefully.

My second concern is that there are many unclear words such as higher, wider, faster, slightly after. These should be quantified as much as possible.

My third concern is about the presentation. The English must be improved, and an experienced English editor should easily pick up words such as "time-serious." Repetitive expressions such as "conductivity are related to salinity" should be deleted. 

Author Response

Comment: It is rather difficult reading this manuscript as the figures provided are not clear. Many important features such as salt fingers and flushing fingers can not be seen(on my screen). Yet, the supplementary films are of much higher quality and show these features clearly. The authors should choose figures more carefully.

Answer: We tried to include many phenomena in a few selected pictures and thus we had to avoid a detailed focusing on each one, e.g., the development of flushing fingers. As you mentioned, some phenomena are better to see in the supplementary videos, thus we direct the reader to them in the paper.

Comment: My second concern is that there are many unclear words such as higher, wider, faster, slightly after. These should be quantified as much as possible.

Answer: Some processes such as time lags are quantified.  Other parameters are difficult to measure in the laboratory and can only be examined using a numerical model. Indeed, some of the main finding of this work, such as mixing mechanisms, are presented without quantitative analysis. Nevertheless, in our opinion, these mechanisms were not considered previously. Our results show new mechanisms for mixing in coastal aquifers.

Comment: My third concern is about the presentation. The English must be improved, and an experienced English editor should easily pick up words such as "time-serious." Repetitive expressions such as "conductivity are related to salinity" should be deleted.

Answer: Following your review, we have sent the paper to English editing.    

Round 2

Reviewer 1 Report

 Lines 124-126: "The sensors were calibrated using several saline solutions with various electrical conductivities, in order to convert voltage to electrical conductivity (Figure 1b), which EC) values. These values are proportional to the salinity extent of the water. inside the system, which varies between 997 and 1027 kg m-3."

R:  I am really astonished with the resistance of the authors to study and report on salinity. The fact that the word "electrode" has ben substituted by enigmatic word "sensor" denotes this, but it is acceptable. Nevertheless other strange terms and concepts have been mistakenly  introduced. Although salinity is the water quality parameter that supports the conclusions of the work, there is neither a bibliographic reference on the subject, nor a value of salinity!!!  In fact, calibration is an operation that establishes the correspondence between the measured signal and the physical parameter it is related with. Therefore the sensors measure conductivities of different salinity solutions "The sensors were calibrated using several saline solutions with various electrical conductivities (Figure 1b) values". The words "...in order to convert voltage to electrical conductivity..." are neither necessary nor correct.  Which are the salinity values of these reference solutions? Also "salinity extent of the water" is strange; why "extent"? Salinity refers to the amount of salts (g) dissolved in 1 kg of water and is conventionally expressed in Practical Salinity (PS) values relative to salinity of standards. Still, the values "997 and 1027 kg m-3" cannot possibly be salinity, they are likely to be density values. 

Author Response

We would like to thank you for his important comment. Indeed, our intention was “density” rather than “salinity”. Accordingly, we have corrected the words throughout the text. Please note specifically the text at lines 120-122 and the supplementary file with the calibration curve of the sensors.

Reviewer 2 Report

The issues I raised were partially addressed, the answers are not always thorough, especially that on the similarities. Despite the work could be futher improved it contains sufficient informations and it may be considered for publication in present form.

Author Response

Thank you for your previous comments.

Indeed, we thought the phenomena observed in this study are worth reporting. However, we agree the information presented in this paper can be more detailed, especially the quantification of the model's parameters and results. We hope to complete it soon in another paper.

Round 3

Reviewer 1 Report

We appreciate the eagerness of the authors to see their manuscript published; nevertheless, their attempt to get there, is done without proper fundamental and practical understanding of what they are measuring, thus leading to unacceptable errors. The fact there are no bibliographic references (requested in a previous revision) concerning this part and topic (Methods, 2.1, Lines 117-126) confirms our judgment. We are led to conclude that they have no basic education and training  in salinity measurement, electroanalytical methods and techniques. Are the authors seeking proper orientation or are they  simply using their own vocabulary, instead of the pertaining terminology and interpretation to express "...measured (R: HOW/EQUIPMENT?) every 30 seconds and collected using Labview software (National Instruments Ltd)"?  It is a shame that so few lines say so much about the lack of expertise of the authors, therefore  the poor quality of the paper. Before eventual resubmission, the authors must ensure that they have fully observed these remarks. There are national and international experts, knowledgeable users and reference literature, where the authors should seek for guidance; search on the Internet, or approach  the firm that supplied the equipment (?) may be a possibility. 

Equation 1-  R: Observe space between symbols of quantities.

L 117-123- "The monitoring setup includes several measurement systems. More than 150 sensors are placed at the backside of the flow tank for in situ voltage values (Figure 1b). The sensors were calibrated using several saline solutions with various electrical conductivities (supplementary material), in order to convert voltage to electrical conductivity (EC) values. These values are proportional to the salinity density extent of the water inside the system, which varies between 997 and 1030 kg m-3. Thus, high and low EC values in a sensor... ". R- SENSORS: are they conductivity sensors or density sensors? They are electrical conductivity sensors. They measure electrical conductivity assessed by sensors placed in conducting solutions, not voltage. The equipment, by application of a constant voltage, reads conductivity, not voltage, thus the sentence "to convert voltage to electrical conductivity" is not correct. The reference solutions are density reference solutions whose EC values are measured (Graph in Supplementary material).  In the text, the scientific notation "kg m-3" is used to represent density units. In the graphs,  density is represented as kg/ m3 (Supplementary material) and conductivity as mS/cm (Supplementary material, Figures 3 and 5); harmonize.

Author Response

Comment: We appreciate the eagerness of the authors to see their manuscript published; nevertheless, their attempt to get there, is done without proper fundamental and practical understanding of what they are measuring, thus leading to unacceptable errors. The fact there are no bibliographic references (requested in a previous revision) concerning this part and topic (Methods, 2.1, Lines 117-126) confirms our judgment. We are led to conclude that they have no basic education and training  in salinity measurement, electroanalytical methods and techniques. Are the authors seeking proper orientation or are they  simply using their own vocabulary, instead of the pertaining terminology and interpretation to express "...measured (R: HOW/EQUIPMENT?) every 30 seconds and collected using Labview software (National Instruments Ltd)"?  It is a shame that so few lines say so much about the lack of expertise of the authors, therefore  the poor quality of the paper. Before eventual resubmission, the authors must ensure that they have fully observed these remarks. There are national and international experts, knowledgeable users and reference literature, where the authors should seek for guidance; search on the Internet, or approach  the firm that supplied the equipment (?) may be a possibility. 

Answer: We have added references of our previous work showing our proper fundamental and practical understanding of what we are measuring (Levanon et al., 2019; Oz et al., 2014, 2015) published in Journal of Hydrology and Geofluids. We changed the wording in lines 117-126 to be similar to our previous publications.

Comment: Equation 1-  R: Observe space between symbols of quantities.

Answer: Fixed

L 117-123- "The monitoring setup includes several measurement systems. More than 150 sensors are placed at the backside of the flow tank for in situ voltage values (Figure 1b). The sensors were calibrated using several saline solutions with various electrical conductivities (supplementary material), in order to convert voltage to electrical conductivity (EC) values. These values are proportional to the salinity density extent of the water inside the system, which varies between 997 and 1030 kg m-3. Thus, high and low EC values in a sensor... ". R- SENSORS: are they conductivity sensors or density sensors? They are electrical conductivity sensors. They measure electrical conductivity assessed by sensors placed in conducting solutions, not voltage. The equipment, by application of a constant voltage, reads conductivity, not voltage, thus the sentence "to convert voltage to electrical conductivity" is not correct. The reference solutions are density reference solutions whose EC values are measured (Graph in Supplementary material).  In the text, the scientific notation "kg m-3" is used to represent density units. In the graphs,  density is represented as kg/ m3 (Supplementary material) and conductivity as mS/cm (Supplementary material, Figures 3 and 5); harmonize.

Answer: Our mistake. We corrected this sentence in our previous round but it somehow appeared again. We fixed it now.

Round 4

Reviewer 1 Report

After several rounds the manuscript is acceptable.

This manuscript is a resubmission of an earlier submission. The following is a list of the peer review reports and author responses from that submission.

Round 1

Reviewer 1 Report

Lines 24-25- Sea tide oscillations….They induce ground

L 30-“  … a significant vertical time-lags”

R: …. significant vertical time-lags

L 114- “168 electrodes placed at the backside of the flow tank for in-situ voltage measurements (Figure 1b)”, and further down.

R: Which electrodes are the authors using? Considering that there are water layers of different salinity, I would imagine practical salinity measurements, i.e. electrical conductivity, which is measured with  conductivity cells in conductivity meters. Conductivity is not Potential and are measured differently; the units for conductivity (Figs 1b,  3 and 5) are not mV!

L 168- Table 1- How have the Salinity values (g(/cm3) been assessed?

Reviewer 2 Report

Dear Authors,

thank you for submitting your manuscript which deals with an interesting topic.
See attached the annotated version, and here are my general comments:

  • The introduction should be written logically and explicitly stating the aims.
  • Results are extremely hard to follow due to the grammar issues and intricate style. Rephrasing of this section is strongly advised.
  • The relevance of your study is missing, e.g. what are the novel outcomes, can you apply your results to any field site?
  • The behaviour of aquifers and aquitards is well known from previous studies and even the effect of layered media.
  • Discussion with other papers is missing, we cannot see the place of your results in the context of this topic.
  • My biggest concern is reproducibility (the haline convection has several time-dependent solutions). The processes and features must be checked by several model runs, at least 5. In this way, your animations would be better (with the same camera position and without people moving around - the animations are not acceptable for publication in the current form).
  • What about the tank size, layer thicknesses and imposed boundary conditions? Are these values reasonable in terms of scale?

The topic seems interesting but the delivery has some flaws and the quality of your manuscript is unfortunately poor.

Reviewer 3 Report

This is a nice, well written paper , which presents the  results of laboratory experiments of tidal effects on the fresh-saline water interface.

My opinion is that these results are of high interest for readers. The results can be used for a better understanding of sea water intrusion processes as well as for model verification.

I suggest to accept  this paper for publication.